# Decoding the Atomic Structure of Ga_2_Te_5_ Pulsed Laser Deposition Films for Memory Applications Using Diffraction and First-Principles Simulations

**DOI:** 10.3390/nano13142137

**Published:** 2023-07-23

**Authors:** Andrey Tverjanovich, Chris J. Benmore, Maxim Khomenko, Anton Sokolov, Daniele Fontanari, Sergei Bereznev, Maria Bokova, Mohammad Kassem, Eugene Bychkov

**Affiliations:** 1Institute of Chemistry, St. Petersburg State University, 198504 St. Petersburg, Russia; andr.tver@yahoo.com; 2X-ray Science Division, Advanced Photon Source, Argonne National Laboratory, Argonne, IL 60439, USA; benmore@anl.gov; 3ILIT RAS-Branch of the FSRC “Crystallography and Photonics” RAS, 140700 Moscow, Russia; khomenkolaser@gmail.com; 4Laboratory of Biophotonics, Tomsk State University, 634050 Tomsk, Russia; 5Laboratoire de Physico-Chimie de l’Atmosphère, Université du Littoral Côte d’Opale, 59140 Dunkerque, France; anton.sokolov@univ-littoral.fr (A.S.); danielefontanari@gmail.com (D.F.); maria.bokova@univ-littoral.fr (M.B.); mohamad.kassem@univ-littoral.fr (M.K.); 6Department of Materials and Environmental Technology, Tallinn University of Technology, 19086 Tallinn, Estonia; sergei.bereznev@taltech.ee

**Keywords:** phase-change materials, synchrotron diffraction, first-principles molecular dynamics

## Abstract

Neuromorphic computing, reconfigurable optical metamaterials that are operational over a wide spectral range, holographic and nonvolatile displays of extremely high resolution, integrated smart photonics, and many other applications need next-generation phase-change materials (PCMs) with better energy efficiency and wider temperature and spectral ranges to increase reliability compared to current flagship PCMs, such as Ge_2_Sb_2_Te_5_ or doped Sb_2_Te. Gallium tellurides are favorable compounds to achieve the necessary requirements because of their higher melting and crystallization temperatures, combined with low switching power and fast switching rate. Ga_2_Te_3_ and non-stoichiometric alloys appear to be atypical PCMs; they are characterized by regular tetrahedral structures and the absence of metavalent bonding. The sp^3^ gallium hybridization in cubic and amorphous Ga_2_Te_3_ is also different from conventional p-bonding in flagship PCMs, raising questions about its phase-change mechanism. Furthermore, gallium tellurides exhibit a number of unexpected and highly unusual phenomena, such as nanotectonic compression and viscosity anomalies just above their melting points. Using high-energy X-ray diffraction, supported by first-principles simulations, we will elucidate the atomic structure of amorphous Ga_2_Te_5_ PLD films, compare it with the crystal structure of tetragonal gallium pentatelluride, and investigate the electrical, optical, and thermal properties of these two materials to assess their potential for memory applications, among others.

## 1. Introduction

Brain-inspired computing [1,2,3], light-controlled (i.e., over a wide spectral range, from visible to THz) or electrically controlled reconfigurable optical metamaterials [4,5,6], high resolution holographic and nonvolatile displays [7,8], integrated photonic circuits [9,10], and many other applications need next-generation phase-change materials (PCMs) with better energy efficiency and wider temperature and spectral ranges to increase reliability compared to current flagship PCMs, such as Ge_2_Sb_2_Te_5_, doped Sb_2_Te, etc. Gallium tellurides seem to be promising candidates to achieve the necessary requirements because of their higher melting and crystallization temperatures, combined with low switching power and fast switching rate [11,12,13]. Ga_2_Te_3_ and non-stoichiometric compositions appear to be atypical PCMs; they are characterized by regular tetrahedral structures, the absence of Peierls distortion in the crystalline phase, and metavalent bonding [14,15,16]. The sp^3^ gallium hybridization in cubic and amorphous Ga_2_Te_3_ [16,17] is also different from conventional p-bonding in flagship PCMs, raising questions about its phase-change mechanism. In addition, gallium tellurides exhibit a number of unexpected and highly unusual phenomena, such as nanotectonic compression [15] and viscosity anomalies just above their melting points [18,19]. Nanotectonic compression involves the simultaneous co-crystallization of the stable ambient and metastable high pressure (HP) forms via heating of glassy g-GaTe_3_ to above the glass transition temperature $T_{g}$. The appearance of metallic HP-polymorphs seems to be beneficial for PCM performance because of the resulting higher optical and electric contrast, accompanied by lower power consumption and the possibility of multilevel writing [16]. The viscosity anomaly for several Ga-Te melts in the vicinity of a sesquitelluride composition [18,19] appears to be more significant than the observed “double kink” in liquid germanium tellurides and other PCM alloys [20,21,22,23], allowing us to distinguish between two contrasting models: (1) fragile-to-strong transition [20,21,22] and (2) incipient liquid–liquid immiscibility [23]. Recent high-energy X-ray [24] and neutron diffraction measurements have shown enhanced small-angle scattering below the scattering vector Q ≲ 0.4 Å^−1^; this is related to the dense metallic liquid droplets in the semiconducting Ga_2_Te_3_ melt and correlates with non-monotonic viscosity behavior, which can be quantitatively described by the Taylor model [25] for two-phase emulsions. The incipient transient immiscibility within a semiconductor–metal transition in liquid telluride PCMs is an interesting topic for further studies. 

The main objectives of the current report are to unravel the atomic structure of amorphous Ga_2_Te_5_ pulsed laser deposition (PLD) film using high-energy X-ray diffraction, supported by first-principles simulations, and to compare it with the crystal structure of tetragonal gallium pentatelluride [26,27]. The PLD technique is often used for PCM growth, as it provides a stoichiometric transfer from the target to the deposited film and achieves a higher deposition rate [28,29]. In contrast to cubic gallium sesquitelluride Ga_2_Te_3_, which is a congruently melting compound in the Ga-Te binary system [30,31,32], tetragonal Ga_2_Te_5_ is stable over a limited temperature range and exhibits peritectic decomposition before melting. Consequently, the relationship between the amorphous material—obtained by the near instantaneous freezing of the highly excited fragments, particles, liquid globules, etc., existing in the laser-induced plasma (plume)—and a metastable crystal is expected to be complex, leaving room for various intermediate configurations and states. Deep insights into the atomic structure and associated electronic, optical, thermal, and other properties are a key for the rational design of next-generation PCMs and new functional materials for use in photovoltaic, thermoelectric, DNA sensing, and energy storage applications [33,34,35,36,37].

## 2. Materials and Methods

### 2.1. Glass and Target Synthesis, Pulsed Laser Deposition

Two different synthesis strategies were applied in efforts to synthesize glassy g-Ga_2_Te_5_ alloys. First, a small sample (300 mg) of high-purity gallium (99.999%, Neyco) and tellurium (99.999%, Cerac) was prepared in a thin-walled silica tube, which was then sealed under vacuum (10^−4^ mbar) and placed in a rocking furnace. The maximum temperature was 1250 K. The synthesized and homogenized sample was rapidly quenched using a salty ice/water mixture. The crystalline sample was synthesized using fast quenching. Second, a two-step synthesis technique was applied for splat quenching. Crystalline Ga_2_Te_5_ was prepared as the intermediate step via a conventional melt quenching technique. A tiny piece of the synthesized material was placed into a silica capillary, heated to ≈1050 K, and kept at this temperature for one hour, with subsequent cooling to 950 K. After a supplementary equilibration step for 30 min, the sample was splat-quenched under an argon atmosphere onto a fused silica plate cooled to ≈80 K. A mostly vitreous sample was obtained via splat-quenching.

The gallium pentatelluride target, Ga_2_Te_5_, for pulsed laser deposition was synthesized in a flat-bottom silica tube with a 25 mm inner diameter. The detailed procedure was described earlier [16], resulting in the final monolithic target with a thickness of 7 mm.

Ga_2_Te_5_ thin films were deposited at room temperature via PLD onto LCD-grade float glass substrates (Kintec Company, Hong Kong) with a substrate diameter of 2” and a thickness of 1.1 mm. A Neocera PLD system equipped with a 248 nm KrF excimer laser (Coherent Compex 102 F) was used for thin film deposition. The laser beam was focused on a ≈5 mm^2^ spot on the surface of the rotating target. The target-to-substrate distance was set at 9 cm, and the pressure in the vacuum chamber was around 3 × 10^−6^ mbar with no background gas pressure. For the preparation of “thick” layers with a thickness of ≈2 μm, 140,000 laser pulses of 200 mJ pulse energy and 10 Hz repetition rate were used. On the other hand, the “thin” layers with a thickness of ≈100 nm were prepared using 7000 laser pulses of 200 mJ pulse energy and 10 Hz repetition rate. A detailed PLD procedure was reported previously [16,38]. The chemical composition of the PLD films was verified via energy dispersive X-ray spectroscopy and found to be consistent with the target composition: 29.3 ± 0.6 at.% Ga and 70.7 ± 2.0 at.% Te.

### 2.2. XRD and DSC Characterization

In situ XRD measurements of a Ga_2_Te_5_ PLD film as a function of temperature have been carried out using a Rigaku Ultima IV diffractometer equipped with a Rigaku SHT-1500 high-temperature platinum camera and sealed Co Kα X-ray tube. The sample was cut from the PLD film on the glass substrate and had dimensions of 9 × 18 mm. The heating rate was 10 K min^−1^ in nitrogen atmosphere. The temperature range was 443 to 653 K with a step of 10 K. After temperature measurements and cooling down the sample, the diffraction pattern was recorded again under ambient conditions. Room-temperature XRD experiments have also been carried out using a Bruker D8 Advance diffractometer (the Cu Kα incident radiation) equipped with a LinxEye detector.

A PLD film powder, removed from the substrate, was used for DSC measurements employing a high precision Netzsch DSC 204 F1 Phoenix instrument (Germany) with μ-sensor. A standard aluminum pan was used for the experiments with a typical heating rate of 10 K min^−1^.

### 2.3. Optical and Electrical Measurements

Optical absorption measurements have been carried out over a wide spectral region from 700 nm to 6 μm. A Shimadzu UV-3600 spectrophotometer was used for experiments in the wavelength range of 700–3200 nm. To cover the far-IR region, a Bruker Tensor FTIR spectrometer was used with the extended IR range up to 25 μm. However, the range was limited by 6 μm due to phonon absorption in the glass substrate. The two instruments had overlapping spectral domains in the range of 2.5 and 3.2 μm.

The electrical conductivity of the samples was measured employing a Hewlett Packard 4194A impedance meter over a frequency range of 100 Hz to 15 MHz. The sample resistance was determined by analyzing the complex impedance plots and was then converted into electrical conductivity using the geometrical factor. Further experimental details can be found elsewhere [15].

### 2.4. High-Energy X-ray Diffraction Measurements

The 6-ID-D beamline at the Advanced Photon Source (Argonne National Laboratory, Chicago, IL, USA) was used for high-energy X-ray diffraction measurements in the top-up mode. The photon energy was 99.9593 keV, and the wavelength was 0.124035 Å. A two-dimensional (2D) setup was used for data collection with a Varex area detector, 2880 × 2880 pixels, and a pixel size of 150 × 150 μm^2^. The sample-to-detector distance was 302.5 mm. A Ga_2_Te_5_ PLD film powder, removed from the substrate, was placed into a silica capillary, which was fixed using a sample holder of the instrument. Cerium dioxide CeO_2_ was used as a calibrant. The 2D diffraction patterns were reduced using the Fit2D software [39]. The measured background intensity of the empty capillary was subtracted, and corrections were made for the different detector geometries and efficiencies, sample self-attenuation, and Compton scattering applying standard procedures [40], providing the X-ray structure factor $S_{X}\left(Q\right)$:(1)$$S_{X}\left(Q\right)=w_{\mathrm{GaGa}}\left(Q\right)S_{\mathrm{GaGa}}\left(Q\right)+w_{\mathrm{GaTe}}\left(Q\right)S_{\mathrm{GaTe}}\left(Q\right)+w_{\mathrm{TeTe}}\left(Q\right)S_{\mathrm{TeTe}}\left(Q\right),$$
where $w_{\mathrm{ij}}\left(Q\right)$ represents Q-dependent X-ray weighting coefficients and $S_{\mathrm{ij}}\left(Q\right)$ represents the Faber–Ziman partial structure factors. 

### 2.5. First-Principles Simulation Details

The Born–Oppenheimer molecular dynamics implemented within the CP2K package [41] was used for the modeling of the diffraction data. The generalized gradient approximation (GGA) and the PBE0 hybrid exchange–correlation functional [42,43] combining the exact Hartree–Fock and DFT approaches were used, providing better agreement with experiments [15,16,38,44,45,46]. The van der Waals dispersion corrections [47] were also employed, improving first-principles molecular dynamics (FPMD) results for telluride systems [48,49]. The applied FPMD technique was similar to previous reports [44,45,46]. The initial atomic configurations for amorphous Ga_2_Te_5_ were created and optimized using the RMC_POT++ code [50] in comparison with the experimental $S_{X}\left(Q\right)$. The cubic simulation box, containing 210 atoms (60 Ga and 150 Te), has a size matching the experimental density. Further optimization was carried out using DFT, applying the molecularly optimized correlation consistent polarized triple-zeta valence basis set along with the norm-conserving relativistic Goedecker–Teter–Hutter-type pseudopotentials [51]. FPMD simulations were performed using a canonical NVT ensemble employing a Nosé–Hoover thermostat [52,53]. The simulation boxes were heated from 300 K to 1500 K using 100 K steps for 20–25 ps each. At the highest temperature, the systems were equilibrated for 30 ps and cooled down to 300 K using the same temperature steps but with a longer simulation time (25–30 ps). Final equilibration and data collection at 300 K were performed for 34 ps. The connectivity and ring statistics were analyzed using the R. I. N. G. S. package [54] and a modified connectivity program [55]. The pyMolDyn code [56] was used for the calculation of microscopic voids and cavities.

## 3. Thermal Properties and Crystallization on Heating

The obtained Ga_2_Te_5_ PLD thin films were found to be amorphous and vitreous according to XRD and DSC measurements. Typical DSC traces of Ga_2_Te_5_ PLD and bulk glassy g-GaTe_3_ are shown in Figure 1e. The endothermic glass transition temperature $T_{g}$ increases with the gallium content from 448 K (g-GaTe_3_, 25 at.% Ga) to 491 K (g-Ga_2_Te_5_, 28.57 at.%). This increase is accompanied by exothermic crystallization. Bulk g-GaTe_3_ shows two intense thermal features, peaking at 492 and 602 K, along with a very weak intermediate effect at 547 K. The 492 K peak corresponds to primary Te crystallization, presenting both the usual trigonal $P3_{1}2$ and high-pressure monoclinic $P2_{1}$ forms. On the other hand, the 602 K feature is associated with the formation of cubic ($F\bar{4}3m$) and high-pressure rhombohedral ($R\bar{3}m$) Ga_2_Te_3_, indicating the occurrence of nanotectonic contraction in a viscous supercooled liquid [15]. 

In contrast, g-Ga_2_Te_5_ PLD film shows a single narrow and intense exothermic effect peaking at 545 K with a crystallization onset at $T_{x}$ ≈ 535 K. The in situ XRD measurements with a typical DSC heating rate of 10 K min^−1^ were used to reveal the nature of crystallizing phase(s). Surprisingly, the first weak Bragg peaks at the scattering angles 2θ = 30.5° and 63.2° (the Co $K_{\alpha }$ incident radiation) have appeared just in the vicinity of $T_{g}$ at ≈483 K, Figure 1a. They correspond to the (002) and (042) reflections of tetragonal Ga_2_Te_5_, I4/m [26], which was reported to be stable only between 673 and 768 K, Figure 1d. These unexpected results suggest the existence of certain Ga_2_Te_5_ motifs in the vitreous PLD film evolving into nano-crystallites on heating in a viscous supercooled liquid. Further crystallization advances in the vicinity of $T_{x}$ when the remaining Bragg peaks of tetragonal Ga_2_Te_5_ become visible and grow over the 513 ≲ T ≲ 653 K temperature range, Figure 1b. In addition to the majority gallium pentatelluride I4/m polymorph, trigonal tellurium, $P3_{1}2$ [57], was also detected as a minority phase, Figure 1c,f. Even after cooling the crystallized sample, the observed phases remain intact, specifically retaining the tetragonal Ga_2_Te_5_, which is metastable at room temperature. 

Even more surprisingly, gallium pentatelluride appears to be perfectly stable after 15 months at room temperature, Appendix A, in contrast to bulk Ga_2_Te_5_, transforming into cubic Ga_2_Te_3_ and trigonal tellurium within several weeks [30]. In other words, a controlled crystallization of the amorphous Ga_2_Te_5_ PLD film yields a high-quality, stable tetragonal crystal promising for photovoltaic, thermoelectric, energy storage, and memory applications [33,34,35,36,37]. On the contrary, the slow cooling or fast quenching of molten Ga_2_Te_5_ gives a polycrystalline mixture of cubic gallium sesquitelluride and trigonal Te, Appendix A, fully consistent with the Ga-Te phase diagram, Figure 1d.

## 4. Electric and Optical Properties

The measured electrical conductivity of bulk crystalline Ga_2_Te_5_ is shown in Figure 2a. In contrast to previously reported results [58], the electrical conductivity follows the Arrhenius relationship over the entire temperature range
(2)$$\sigma =\sigma_{0}\exp\left(-E_{a}/k_{B}T\right),$$
where $\sigma_{0}$ is the pre-exponential factor, $E_{a}$ is the conductivity activation energy, and $k_{B}$ and T have their usual meanings. Nevertheless, the derived activation energy $E_{a}$ = 0.227 ± 0.003 eV is identical to that for intrinsic conductivity in tetragonal single-crystal Ga_2_Te_5_, measured in the direction perpendicular to the c-axis [58]. The conductivity pre-factor, $\sigma_{0}$ = 17 ± 3 S cm^−1^, was situated at the lower limit of the electronic transport regime over the extended states [59]. This indicates that $2E_{a}$ = 0.45 eV is approximately the electrical bandgap.

The electrical conductivity for glassy g-Ga_2_Te_5_ was obtained via the interpolation of the available data for amorphous and glassy ${\mathrm{Ga}}_{x}{\mathrm{Te}}_{1-x}$ alloys, where 0 ≤ x ≤ 0.4 [15,60,61], as shown in Figure 2b. The room-temperature conductivity appears to be a decreasing exponential function of the gallium content x:(3)$$\sigma_{298}\left(x\right)=\sigma_{298}\left(0\right)\exp\left(ax\right)\;,$$
where $\sigma_{298}\left(0\right)$ is the conductivity of amorphous Te and a < 0 is a constant. In other words, the electronic conductivity of disordered Ga-Te materials primarily relies on the tellurium concentration. The interpolated g-Ga_2_Te_5_ conductivity is lower by two orders of magnitude compared to the $\sigma_{298}$ value of bulk crystalline pentatelluride. The estimated conductivity activation energy, $E_{a}$ = 0.41 eV, and the pre-exponential factor, $\sigma_{0}$ ≈ 220 S cm^−1^, suggest the electrical bandgap of the glassy polymorph is about 0.8 eV.

Figure 3 shows the optical properties of Ga_2_Te_5_ PLD films. The absorption measurements reveal a fundamental optical absorption edge below the incident wavelength λ ≲ 1.2 μm accompanied by distinct interference fringes. These fringes indicate a homogeneous nature and uniform thickness of the PLD film. The observed fringes allow both the refractive index $n_{R}$ and the film thickness to be calculated using the Swanepoel method [62]. Moreover, due to the well-defined interference across the spectral range and the considerable thickness of the PLD film (2.7 μm), it also becomes possible to estimate the refractive index dispersion $n_{R}\left(\lambda \right)$.

Two approaches are usually applied to represent the refractive index dispersion [63]. The Cauchy approximation of the derived data $n_{R}\left(\lambda \right)$ is given in Figure 3c, as follows:(4)$$n_{R}\left(\lambda \right)=A+\frac{B}{\lambda^{2}}+\frac{C}{\lambda^{4}},$$
where A, B, and C are constants which are characteristic of any given material. Since the Cauchy equation is inappropriate in a region of anomalous dispersion [63], the Sellmeier approach is often used, considering the existence in an optical material of dipole oscillators with a resonance frequency $\upsilon_{0}$:(5)$$n_{R}{\left(\lambda \right)}^{2}=1+\frac{A\lambda^{2}}{\lambda^{2}-\lambda_{0}^{2}},$$
where A and $\lambda_{0}=c/\upsilon_{0}$ are two characteristic constants, and c is the speed of light. Usually, the Sellmeier equation is written with a series of terms to account for different resonance frequencies over an extended domain, that is, $\upsilon_{0}$, $\upsilon_{1}$, etc.:(6)$$n_{R}{\left(\lambda \right)}^{2}=1+\frac{A_{0}\lambda^{2}}{\lambda^{2}-\lambda_{0}^{2}}+\frac{A_{1}\lambda^{2}}{\lambda^{2}-\lambda_{1}^{2}}+\ldots$$

The Sellmeier coefficients $A_{i}$ and $\lambda_{i}$ allow the normal dispersion of optical glasses to be calculated over a wide spectral range. In our case, we were limited to the original Sellmeier equation (5) with the following coefficients: $A_{0}$ = 3.5017 and $\lambda_{0}$ = 0.3992 (Figure 3d). Due to an insufficient spectral range and experimental uncertainty, the higher-order terms in the refractive index dispersion could not be accessed. Nevertheless, despite this limitation, the two approaches describe the derived $n_{R}\left(\lambda \right)$ values reasonably well.

The optical absorption results were also used to calculate the optical bandgap $E_{g}$ applying the Tauc relation [64]:(7)$$\alpha =\frac{A{\left(h\nu -E_{g}\right)}^{2}}{h\nu },$$
where α is the absorption coefficient, hν the photon energy, and A ≈ 10^5^ cm^−1^ eV^−1^ is a constant.

As expected, the derived bandgap $E_{g}$ = 0.98 ± 0.02 eV, Figure 3b, for glassy gallium pentatelluride was found to be smaller than that for g-Ga_2_Te_3_, which was 1.20 eV [16], supporting the predominant role of the tellurium content on electronic and optical properties of Ga-Te alloys. Simultaneously, the optical $E_{g}$ appears to be comparable with the electrical counterpart, $2E_{a}$ = 0.82 eV.

The thermally annealed and crystallized Ga_2_Te_5_ PLD film exhibits more complicated optical absorption, Figure 4. The absorbance below λ ≲ 1 μm shows a distinct blue-shift, while the low-energy absorbance becomes more intense and mostly loses interference fringes indicating less homogeneous material in both the chemical composition and thickness. Taking into account the presence of (nano)crystallites in the annealed PLD film, additional scattering corrections were applied simultaneously with the usual reflection corrections. The refractive index $n_{R}\left(\lambda \right)$ of g-Ga_2_Te_5_ was used for these calculations.

The Mie theory of light scattering for turbidity τ measurements and the wavelength exponent $\lambda^{-\chi }$ were employed for the scattering corrections [65,66]:(8)$$\tau =l^{-1}\ln\left(I_{0}/I\right),$$
where l is the scattering path length, and $I_{0}$ and I are the intensities of the incident and transmitted beam, respectively. The turbidity depends on several parameters:(9)$$\tau =B_{s}N_{p}V_{p}^{2}\lambda^{-\chi },$$
where $B_{s}$ is the scattering coefficient, $N_{p}$ is the particle number density, $V_{p}$ is the average volume of the particle, and χ is the wavelength exponent. Combining Equations (8) and (9), one obtains
(10)$$D=K_{s}\lambda^{-\chi }$$
where $D=\log\left(I_{0}/I\right)$ is the optical density and $K_{s}$ is a constant. The values of $K_{s}$ and χ were obtained by plotting lnD vs. lnλ (Appendix A), which allows both the turbidity τ(λ) and χ to be determined and the scattering corrections to be calculated. The theoretical Heller wavelength exponent $\chi_{0}$ [65] yields the average particle size $\langle r_{p}\rangle$, which was found to be $\langle r_{p}\rangle$ ≈ 110 nm for c-Ga_2_Te_5_, Appendix A. The derived $\langle r_{p}\rangle$ value is consistent with the size of crystallites, obtained from the XRD linewidth Γ, yielding $\langle r_{p}^{\mathrm{XRD}}\rangle =2\pi /\Gamma$ > 50 nm. The final absorbance corrected for reflection and scattering is shown in Figure 4a.

The derived optical absorption coefficient α, presented in Figure 4b, exhibits two optical processes above and below the incident photon energy hν ≈ 1.3 eV. Basically, the overall shape of the absorption coefficient α is reminiscent of the behavior observed in materials such as silicon or carbon [67,68]. This shape is typically associated with direct optical transition at higher photon energies hν and indirect optical absorption at lower hν [68,69,70,71,72]. Assuming direct optical transition in crystallized Ga_2_Te_5_ above 1.3 eV,
(11)$$\alpha h\nu =A{\left(h\nu -E_{g}^{d}\right)}^{1/2},$$
the direct optical bandgap was found to be $E_{g}^{d}$ = 1.36 ± 0.03 eV, Figure 4c. The constant A in Equation (11) is given by $A\approx \left[e^{2}/\left(n_{R}ch^{2}m_{e}^{*}\right)\right]{\left(2m^{*}\right)}^{3/2}$, where e is the electron charge and $m^{*}$ is a reduced electron and hole effective mass [69].

The optical absorption plotted as $\sqrt{\alpha h\nu }$ vs. photon energy hν, Equation (7), yields the indirect optical bandgap $E_{g}^{i}$ = 0.40 ± 0.03 eV, Figure 4d. The derived value is consistent with the electrical bandgap of c-Ga_2_Te_5_, $2E_{a}$ = 0.45 eV, Figure 2a.

The experimental data for tetragonal Ga_2_Te_5_, obtained using the conductivity and Hall effect measurements, are strongly anisotropic and changing over a wide range between 0.46 and 1.79 eV [58]. The calculated $E_{g}$ values are also variable, 0.86 ≤ $E_{g}$ ≤ 1.7 eV [72,73], depending on the applied simulation method. Nevertheless, the results of electrical and optical measurements show a reasonable contrast between amorphous (SET) and crystalline (RESET) states for Ga_2_Te_5_.

## 5. High-Energy X-Ray Diffraction

The high-energy X-ray diffraction data in Q-space are shown in Figure 5. In contrast to fast cooled Ga_2_Te_5_ in a thin-walled silica tube, mostly consisting of cubic Ga_2_Te_3_ and trigonal tellurium with some vitreous fraction (Appendix A), the splat-quenching of tiny Ga_2_Te_5_ droplets yields an essentially glassy material. However, this glassy material is accompanied by non-negligible nanocrystals of cubic gallium sesquisulfide, as seen in Figure 5a. The spontaneous Ga_2_Te_3_ crystallization is consistent with the Ga-Te phase diagram, Figure 1d, related to peritectic decomposition of Ga_2_Te_5_ above 768 K. The obtained Ga_2_Te_5_ PLD films are fully vitreous with a distinct glass transition temperature at 491 K, Figure 1e. The X-ray structure factor $S_{X}\left(Q\right)$ of g-Ga_2_Te_5_ PLD appears to be intermediate between bulk glassy GaTe_4_ and Ga_2_Te_3_ PLD film, Figure 5b, suggesting structural similarities and revealing a systematic evolution of vitreous ${\mathrm{Ga}}_{x}{\mathrm{Te}}_{1-x}$ materials with increasing gallium content x independently of preparation techniques.

In particular, we observe an emerging and growing first sharp diffraction peak (FSDP), also shifting to a lower Q with increasing x from $Q_{0}$ = 0.94 ± 0.01 Å^−1^ (GaTe_4_, x = 0.2) to 0.86 ± 0.01 Å^−1^ (Ga_2_Te_3_, x = 0.4), Figure 5c. The isolated FSDPs were obtained using the subtraction procedure [74,75]. The FSDP systematics (position $Q_{0}$ and amplitude $A_{0}$) reveals monotonic and nearly linear trends as a function of x, Appendix A, for both bulk glasses (0.17 ≤ x ≤ 0.25) and vitreous PLD films (0.2857 ≤ x ≤ 0.40). The observed trends suggest that there are structural similarities on the short- and intermediate-range scale.

Distinct high-Q oscillations, clearly visible over the extended Q-range for the Ga_2_Te_5_ interference function $Q[S_{X}\left(Q\right)-1]$, Figure 5d, enable high real-space resolution for atomic pair distribution $g_{X}\left(r\right)$ and total correlation $T_{X}\left(r\right)$ functions after the usual Fourier transform:(12)$$T_{X}\left(r\right)=4\pi \rho_{0}r+\frac{2}{\pi }\int_{0}^{Q_{\max}}Q\left[S_{X}\left(Q\right)-1\right]\sin QrM\left(Q\right)dQ,$$
where $\rho_{0}$ is the experimental number density, M(Q) the Lorch window function, and $Q_{\max}$ = 30 Å^−1^.

The derived $T_{X}\left(r\right)$ for g-Ga_2_Te_5_ PLD film is shown in Figure 6. The asymmetric feature between 2.4 and 3.2 Å corresponds to Ga-Te and Te-Te nearest neighbors (NN). Gaussian fitting of the experimental data (Table 1) yields tetrahedral gallium coordination at a distance of 2.64 Å, consistent with the known Ga-Te coordination numbers and NN distances in crystalline and glassy gallium tellurides [14,15,16,26,27,30,76,77,78]. On the contrary, the Te-Te atomic pairs in glassy Ga_2_Te_5_ are markedly shorter (2.80 Å) than those in tetragonal gallium pentatelluride (3.027 Å) [26]. Nevertheless, the Te-Te NN distance of 2.80 Å is typical for amorphous and trigonal tellurium [57,79] and Te-rich binary and ternary glasses [15,77,78,80]. The partial $T_{\mathrm{ij}}\left(r\right)$ correlation functions for tetragonal gallium pentatelluride are compared in Figure 6b with an experimental $T_{X}\left(r\right)$ for g-Ga_2_Te_5_. We note both similarities and differences for the two materials.

The crystalline counterpart is stable over a narrow temperature range from 673 to 768 K [30,31,32], as shown in Figure 1d. In contrast to layered compounds like Al_2_Te_5_ and In_2_Te_5_ [27], gallium pentatelluride has a 3D structure consisting of infinite chains, parallel to the *c* axis. The chains are formed by edge-sharing ES-GaTe(II)_4_ tetrahedra, Figure 6c. Additionally, every four ES-GaTe(II)_4_ entities from neighboring chains are linked together by Te(I) species, located at the center of squares, formed by Te(II).

These square-planar Te_5_ units (crosses) are mostly absent in glassy Ga_2_Te_5_, as evidenced by the significantly lower Te-Te NN coordination number, $N_{\mathrm{Te}-\mathrm{Te}}^{\mathrm{PLD}}$ = 1.01 ± 0.08 (Table 1), compared to the crystal counterpart, $N_{\mathrm{Te}-\mathrm{Te}}^{I4/m}$= 1.6 = ⅕ × 4 + ⅘ × 1, which represents the average Te-Te coordination for the Te(I) and Te(II) species, Figure 6c. On the other hand, the existence of ES-GaTe_4_ tetrahedra in the glass network is indicated by the presence of a weak shoulder at 3.39 ± 0.02 Å for the asymmetric second neighbor peak, centered at ≈4.3 Å. The short Ga-Ga second neighbor correlations, characteristic of ES-units in tetragonal Ga_2_Te_5_, are located at 3.424 Å, Figure 6b. However, due to a weak average Ga-Ga weighting factor, $\langle w_{\mathrm{GaGa}}^{X}\left(Q\right)\rangle$ = 0.02964 vs. $\langle w_{\mathrm{GaTe}}^{X}\left(Q\right)\rangle$ = 0.28405 or $\langle w_{\mathrm{TeTe}}^{X}\left(Q\right)\rangle$ = 0.68630, the amplitude of this feature is small. The weak average Ga-Ga weighting factor, as well as the truncation ripples, related to a finite Q-range of the Fourier transform, enable only a rough estimation of $N_{\mathrm{Ga}-\mathrm{Ga}}^{\mathrm{PLD}}$ = 1.2 ± 0.4, compared to $N_{\mathrm{Ga}-\mathrm{Ga}}^{I4/m}$ = 2. A deep insight into the atomic structure of vitreous Ga_2_Te_5_ PLD films yields first-principles molecular dynamics.

## 6. First-Principles Molecular Dynamics

Simulated FPMD X-ray structure factor $S_{X}\left(Q\right)$ and pair-distribution function $g_{X}\left(r\right)$ for glassy Ga_2_Te_5_ in comparison with experimental results are shown in Figure 7a,b. We note that the GGA approximation with hybrid PBE0 functional describes the experimental data well, as it was reported earlier [15,16,44,45,46]. The positions and amplitudes of the diffraction features in both Q- and r-space are reproduced.

The calculated partial structure factors $S_{\mathrm{ij}}\left(Q\right)$ are displayed in Figure 7c. As expected, the main contribution to the FSDP comes from the Ga-Ga partial $S_{\mathrm{GaGa}}\left(Q\right)$. The simulated $g_{\mathrm{ij}}\left(r\right)$, Figure 7d, reveal complicated short- and intermediate-range orders.

The asymmetric Ga-Te NN correlations appear at 2.62 Å and suggest at least two contributions with slightly different bond lengths. The Ga-Te coordination number is consistent with the experiment, $N_{\mathrm{Ga}-\mathrm{Te}}^{\mathrm{FPMD}}$ = 3.97, Table 1, assuming a tetrahedral gallium local environment. In addition to homopolar Te-Te bonds at 2.80 Å, a weak Ga-Ga NN feature at 2.42 Å was also found. The amplitude of this peak is too small to be observed experimentally, Figure 7b. The Ga-Ga second neighbors between 3 and 4.5 Å have a bimodal distribution. The shoulder at ≈3.35 Å indicates the ES-units, while the main contribution at 3.92 Å is related to corner-sharing CS-entities. Consequently, the fraction of ES-GaTe_4_ in the glassy Ga_2_Te_5_, $f_{\mathrm{ES}}^{\mathrm{FPMD}}$ = 0.45, is significantly lower than that in tetragonal polymorph, $f_{\mathrm{ES}}^{I4/m}$ = 1, indicating that only ES-GaTe_4_ are present in the crystalline counterpart. The experimental value, $f_{\mathrm{ES}}^{\mathrm{PLD}}$ = 0.6 ± 0.2, is reasonably consistent with the FPMD result. Basically, the experimental and FPMD structural parameters were found to be similar or identical, Table 1.

The Ga and Te local coordination distributions are presented in Figure 8. The tetrahedral gallium coordination $\mathrm{Ga}\left({\mathrm{Te}}_{4-m}{\mathrm{Ga}}_{m}\right)$ contains negligible number of Ga-Ga homopolar pairs (m = 1). In contrast, tellurium has multiple coordination environments $\mathrm{Te}\left({\mathrm{Ga}}_{n-m}{\mathrm{Te}}_{m}\right)$, 1 ≤ n ≤ 4, but only two-fold Te_2F_ (50.5%) and three-fold Te_3F_ (47.6%) coordinated species are the most abundant. The tellurium forms reveal a large variability in Te-Te bonds, 0 ≤ m ≤ 4, from pure heteropolar Te-Ga coordination (m = 0) to fully homopolar environment (m = n). We should, however, note a small fraction of Te_4F_ species (1.87%) and a negligible number of Te_5_ square-planar entities (0.23%), the only form of tellurium subnetwork in tetragonal gallium pentatelluride, Figure 6c. This result is coherent with the reduced Te-Te coordination number $N_{\mathrm{Te}-\mathrm{Te}}^{\mathrm{PLD}}≅N_{\mathrm{Te}-\mathrm{Te}}^{\mathrm{FPMD}}≅1$, Table 1.

The geometry of GaTe_4_ units yields either Te-Ga-Te bond angles or the orientational order parameter q [81,82]. Figure 9a,c show the calculated $B_{\mathrm{TeGaTe}}\left(\theta \right)/\sin\theta$ bond angle distribution for g-Ga_2_Te_5_ in comparison with tetragonal Ga_2_Te_5_ and cubic Ga_2_Te_3_, respectively. We note a broad and slightly asymmetric $B_{\mathrm{TeGaTe}}\left(\theta \right)/\sin\theta$ function, centered at 103.3 ± 0.3°, for the PLD film (Appendix A). The Te-Ga-Te angles in the two crystalline references, characterizing both distorted ES-GaTe_4_ tetrahedra in gallium pentatelluride and regular CS-units in cubic sesquitelluride, are located within the glassy envelope but do not reproduce it via simple broadening. Nevertheless, the tetrahedral geometry in tetragonal Ga_2_Te_5_ seems to be closer to that in the glass.

The connectivity of GaTe_4_ tetrahedra is given by the Ga-Te-Ga triplets or the respective $B_{\mathrm{GaTeGa}}\left(\theta \right)/\sin\theta$ distributions, Figure 9b,d. The difference in connectivity between g-Ga_2_Te_5_ and the crystalline structures is even more significant but a remote resemblance to the connected ES-entities in the tetragonal crystal still exists.

The order parameter q [81,82] is often used to evaluate the polyhedral topology and distinguish between tetrahedral and non-tetrahedral local geometry of four-fold coordinated GaTe_4_ entities
(13)$$q=1-\frac{3}{8}\sum_{j=1}^{3}\sum_{k=j+1}^{4}{\left(\cos\psi_{jk}+\frac{1}{3}\right)}^{2},$$
where $\psi_{jk}$ is the Te-Ga-Te angle of the central gallium atom with its nearest Te neighbors j and k. The average value of q changes between 0 for an ideal gas and q = 1 for a regular tetrahedral network. The P(q) probability distribution function is shown in Figure 10a. Asymmetric P(q) is peaked at q = 0.93 and decreases sharply both ways to high and low q. Usually, the tetrahedral limit is set at q ≥ 0.8 [44,83]. The P(q) integration within these limitations shows that 97% of GaTe_4_ units belong to the tetrahedral geometry. The remaining entities (3%) are likely defect octahedral species GaTe_4_ with two missing Te neighbors around the central Ga atom, characterized by 0.4 < q < 0.8. The regular defect octahedron ($\psi_{1}=\pi$ and $\psi_{2}=\pi /2$) has q = ⅝. The obtained results differ drastically from flagship PCM (GeTe, Ge_2_Sb_2_Te_5_), which show a significant fraction of defect octahedral sites and a minority of tetrahedral structural units (≲40%) [28,83,84,85,86,87]. The exact proportion varies depending on experimental technique, that is, EXAFS or diffraction, and the exchange–correlation functional for FPMD simulations.

Two-fold and three-fold coordinated tellurium can explain the asymmetric shape of the Ga-Te NN peak. The calculated Ga-Te_2F_ and Ga-Te_3F_ distances are presented in Figure 10b. The two distributions are broad and asymmetric but have slightly different maxima. The Ga-Te_3F_ bonds are longer (a broad maximum at 2.71 ± 0.03 Å) compared to Ga-Te_2F_, peaked at 2.63 ± 0.02 Å. A similar difference, $\Delta r_{3F-2F}$ = 0.12 ± 0.01 Å, was reported in monoclinic Ga_2_S_3_ [88] with an ordered distribution of gallium vacancies. The ratio of Ga-Te_3F_ to Ga-Te_2F_ bond populations, $r=\left[{\mathrm{Ga}-\mathrm{Te}}_{3F}\right]/\left[{\mathrm{Ga}-\mathrm{Te}}_{2F}\right]$ = 1.75, was also found to be similar to the expected stoichiometric ratio $r_{0}$ = 2 for tetrahedral Ga species, which possess the formal oxidation state Ga^3+^ and correspond to the Ga_2_Te_3_ alloy composition.

The connectivity analysis shows that all Ga and Te species are connected. The analysis of Te-Te connectivity reveals a different size of ${\mathrm{Te}}_{k}$ fragments, Figure 11a. Tellurium monomers (k = 1), that is, Te atoms with only heteropolar Te-Ga bonds, and dimers (k = 2) represent a relative majority, 55%, of all ${\mathrm{Te}}_{k}$ fragments. The remaining fragments can be divided into two groups: (i) 3 ≤ k ≤ 6 (see the inset in Figure 11a) and (ii) oligomeric chains, k = 15, for the used 210-atom simulation box. Group (i) represents the remnants of square-planar Te_5_ units in tetragonal Ga_2_Te_5_ (the inset in Figure 11b), also confirmed by bond angle distribution $B_{\mathrm{TeTeTe}}\left(\theta \right)/\sin\theta$. Group (ii) is similar to chains in trigonal tellurium, $P3_{1}2$ [57], supported by a contribution at about $\theta_{\mathrm{TeTeTe}}$ ≈ 103°, Figure 11b. The presence of two distinct groups is likely a consequence of the limited thermal stability of tetragonal gallium pentatelluride and peritectic reaction Ga_2_Te_5_ ⇄ Ga_2_Te_3_ +2Te above 768 K.

The intermediate-range order in glassy and amorphous materials is often described by ring statistics, that is, by the population of ${\mathrm{Ga}}_{p}{\mathrm{Te}}_{q}$ rings in the case of gallium tellurides. The ring population $R_{c}\left(p\;+q\right)$ [54] in glassy Ga_2_Te_5_ (this work) and Ga_2_Te_3_ [16] PLD films in comparison with crystalline references (tetragonal Ga_2_Te_5_ [26], cubic [17], and rhombohedral [89] Ga_2_Te_3_) is shown in Figure 12. The $R_{c}\left(p\;+q\right)$ population was found to be different for the two Ga_2_Te_5_ forms. The dominant eight-membered rings in tetragonal polymorphs are hardly populated in the PLD film. On the contrary, the most populated p +q = 5 entities in g-Ga_2_Te_5_ are absent in c-Ga_2_Te_5_. The peritectic nature of tetragonal crystal seems to be related to this difference.

Gallium sesquitelluride Ga_2_Te_3_ is a congruently melting compound, Figure 1d. As it was reported earlier [16], the ring statistics in g-Ga_2_Te_3_ represent a disordered mixture of $R_{c}\left(p\;+q\right)$ for the ambient and high-pressure polymorphs, Figure 12b,d, related to nanotectonic contraction in a viscous supercooled melt.

Microscopic voids and cavities in amorphous Ga-Te alloys, obtained using the Dirichlet–Voronoi tessellation [56], are displayed in Figure 13b. The fraction of voids $V_{c}$, normalized to the volume of the FPMD simulation box, was found to be nearly invariant, 27 ≤ $V_{c}\left(x\right)$ ≤ 29%, over the gallium content x between 0.20 (bulk g-GaTe_4_) and 0.40 (PLD g-Ga_2_Te_3_). This is coherent with a small change in the number density, ≈2% over the same composition range. Typical cavity radius varies between 0.2 and 4 Å, slightly increasing with x, Figure 13a.

The total electronic density of states (eDOS) is shown in Figure 14 and appears to be typical for glassy and crystalline chalcogenides [15,16,59,90,91,92]. The valence band (VB) consists of three sub-bands between the Fermi level $E_{F}$ and −16 eV. The upper part, roughly centered at −3 eV, mostly consists of Te 5p and Ga 4p states and also includes non-negligible d-electron contributions, as it revealed by the eDOS projections (pDOS) on Ga and Te atomic pseudo-wave functions. The middle-energy sub-band, centered at −8 eV, essentially contains Te 5p and Ga 4s electron states, while the lower part, peaked at −13 eV, has an s-character, populated by Te 5s electrons together with Ga s-, p- and d-electron contributions. The derived eDOS and pDOS are similar to those in g-Ga_2_Te_3_ PLD film [16] and suggest sp^3^ gallium hybridization also has a d-electron contribution.

The inverse participation ratio IPR [93,94] allows for localized (large IPR → 1) and extended electron states (small IPR ≈ $N^{-1}$, where N is the number of atoms in the simulation box) to be distinguished
(14)$$\mathrm{IPR}=\frac{\int dr{\left|\psi \left(r\right)\right|}^{4}}{{\left(\int dr{\left|\psi \left(r\right)\right|}^{2}\right)}^{2}},$$
where ψ(r) is a single-particle Kohn–Sham eigenfunction. The calculated IPRs, derived using the projections of ψ(r) onto an atomic basis set and the atomic orbital coefficients, are shown in Figure 14a, plotted together with the eDOS. As it was reported earlier [16,46], a higher electron localization appears at the band tails (the top of the valence and the bottom of the conduction bands), consistent with the theories of disordered semiconductors [59]. The remaining electron states in the vicinity of the bandgap are delocalized (extended). Deeper states of the lower-energy sub-bands, participating in the covalent bonding, are localized even more strongly.

The derived GGA/PBE0 bandgap, $E_{g}^{\mathrm{PBE}0}$ = 0.80 eV, appears to be smaller than the experimental optical bandgap, $E_{g}$ = 0.98 eV, Figure 3b, but nearly identical to the interpolated electrical counterpart, $2E_{a}$ = 0.82 eV, Figure 2a. The main contribution of the Te 5p electron states to the upper part of the valence band and at the bottom of the conduction band is also consistent with a dominant role played by Te on the electronic conductivity of Ga-Te alloys, Figure 2b.

## 7. Conclusions

Pulsed laser deposition allows for homogeneous and uniform Ga_2_Te_5_ films to be obtained, showing a distinct glass transition at 491 K and accompanied by strong crystallization peaked at 545 K. Thermal annealing of the PLD film with a DSC heating rate and carried out below the stability limits of tetragonal Ga_2_Te_5_ yields a high-quality and long-living (for at least 15 months) tetragonal polymorph that is thermodynamically metastable under ambient conditions. Amorphous and crystalline Ga_2_Te_5_ forms show reasonably high electric contrast (two orders of magnitude at room temperature) and distinctly different optical band gaps, $E_{g}$ = 0.98 eV for g-Ga_2_Te_5_ and indirect optical bandgap $E_{g}^{i}$ = 0.40 eV for c-Ga_2_Te_5_. Consequently, gallium pentatelluride can be used for memory applications as well as for photovoltaics, quantum-dots-based DNA sensors, and for the preparation of atomically thin layers via the controlled crystallization of amorphous thin films.

The local coordination of tetrahedral gallium is common for these two forms; however, the intermediate-range order and tellurium subnetwork are drastically different. Square-planar Te_5_ units (crosses) connecting edge-sharing Ge-Te chains in tetragonal Ga_2_Te_5_, which are formed by ES-GaTe_4_ tetrahedra, do not survive in the glassy polymorph, leaving ${\mathrm{Te}}_{k}$ remnants, 3 ≤ k ≤ 6, originating from the Te_5_ entities and oligomeric tellurium chains similar to those in trigonal Te. Quasi-1D edge-sharing Ga-Te chains lose their exclusive structural signature, characteristic of tetragonal polymorph, transforming into 2D and 3D structural patches of edge- and corner-sharing GaTe_4_ tetrahedra. The simulated electronic density of states is consistent with the experimental optical and conductivity results and reveals a predominant role of the Te 5p electron states for electronic properties of gallium pentatelluride.

## Figures and Tables

**Figure 1 nanomaterials-13-02137-f001:**
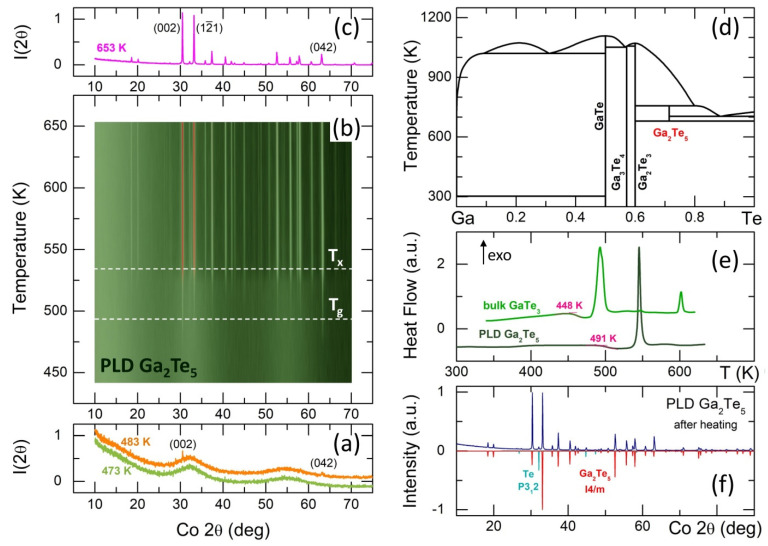
Thermal properties of Ga_2_Te_5_ PLD films and crystallization on heating. In situ diffraction measurements (**a**) at 473 and 483 K, (**b**) between 443 and 653 K, (**c**) at 653 K; (**d**) Ga−Te phase diagram [30,31,32]; (**e**) typical DSC traces for Ga_2_Te_5_ PLD and bulk glassy g−GaTe_3_, and the glass transition temperatures are also indicated; (**f**) identification of the crystallized phases in Ga_2_Te_5_ PLD film after cooling from 653 K to room temperature. Trigonal tellurium, $P3_{1}2$ [57], and tetragonal Ga_2_Te_5_, I4/m [26], were found.

**Figure 2 nanomaterials-13-02137-f002:**
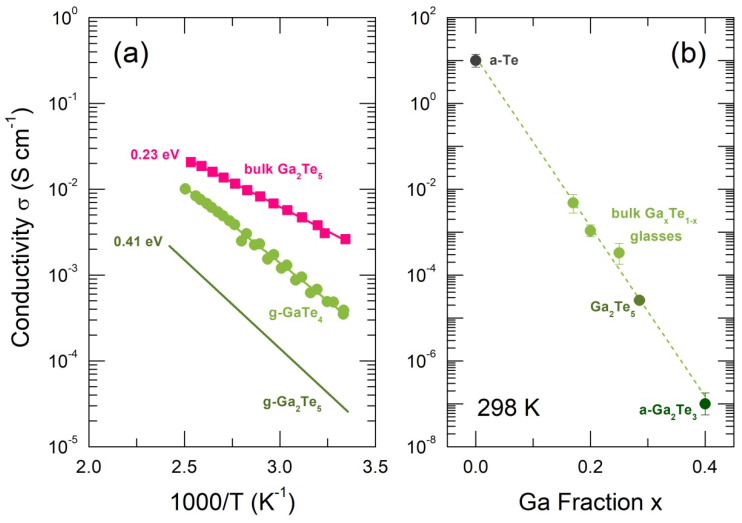
Electrical properties of ${\mathrm{Ga}}_{x}{\mathrm{Te}}_{1-x}$ amorphous, glassy, and crystalline materials: (**a**) electrical conductivity of bulk crystalline Ga_2_Te_5_ (this work), bulk glassy g−GaTe_4_ [15], and interpolated conductivity for g−Ga_2_Te_5_, and the derived activation energies $E_{a}$ are also indicated; (**b**) room-temperature conductivity for amorphous a−Te thin film, corresponding to band-to-band electronic transport [60], bulk Ga−Te glasses [15], and amorphous a−Ga_2_Te_3_ [61]; the interpolated value for g−Ga_2_Te_5_ is also shown.

**Figure 3 nanomaterials-13-02137-f003:**
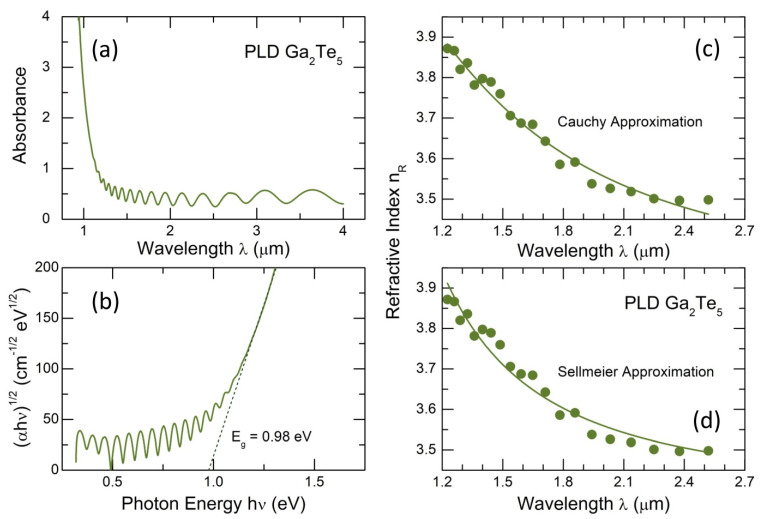
Optical properties of Ga_2_Te_5_ PLD films: (**a**) optical absorbance with interference fringes as a function of the incident wavelength λ; (**b**) the Tauc plot, $\sqrt{\alpha h\nu }$ vs. photon energy hν, where α is the absorption coefficient, yielding the optical band gap $E_{g}$; (**c**) the Cauchy and (**d**) Sellmeier approximations [63] describing the dispersion of the refractive index $n_{R}$ as a function of λ. See the text for further details.

**Figure 4 nanomaterials-13-02137-f004:**
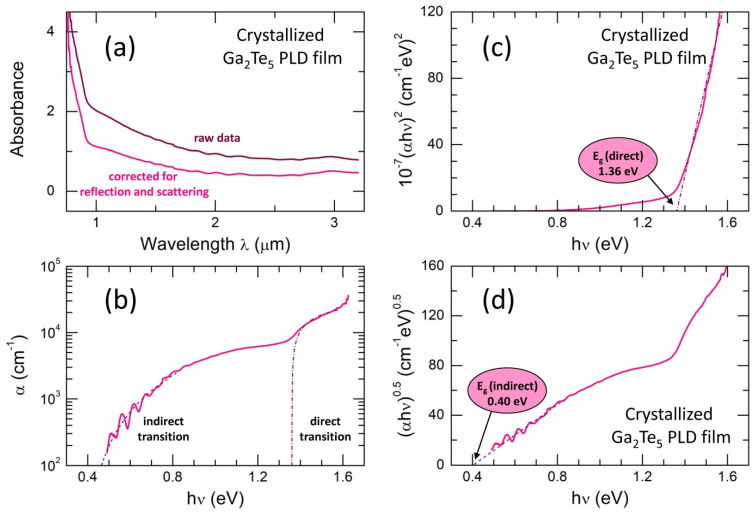
Optical properties of crystallized Ga_2_Te_5_ PLD film: (**a**) raw absorbance data and absorbance corrected for reflection and scattering; (**b**) absorption coefficient α and the analysis results for direct (the dash-dotted line) and indirect (the dashed line) optical transitions; (**c**) data analysis suggesting direct optical transition; the derived direct optical gap $E_{g}^{d}$ is also indicated; (**d**) data analysis assuming indirect optical transition; the derived indirect optical gap $E_{g}^{i}$ is indicated. See the text for further details.

**Figure 5 nanomaterials-13-02137-f005:**
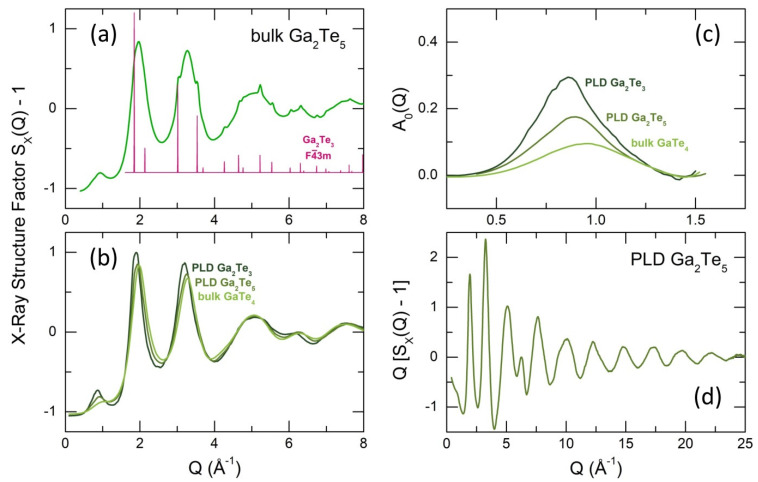
Diffraction data in Q-space: the X-ray structure factor $S_{X}\left(Q\right)$ of (**a**) bulk splat-quenched Ga_2_Te_5_ and (**b**) Ga_2_Te_3_, Ga_2_Te_5_ PLD films and bulk glassy g−GaTe_4_ over a limited Q -range; (**c**) isolated first sharp diffraction peaks (FSDP) for Ga_2_Te_3_ and Ga_2_Te_5_ PLD films and g−GaTe_4_; (**d**) the interference function $Q[S_{X}\left(Q\right)-1]$ for Ga_2_Te_5_ PLD film over the extended Q −range. The Bragg peaks for cubic Ga_2_Te_3_ (space group $F\bar{4}3m$ ) are also shown in (**a**).

**Figure 6 nanomaterials-13-02137-f006:**
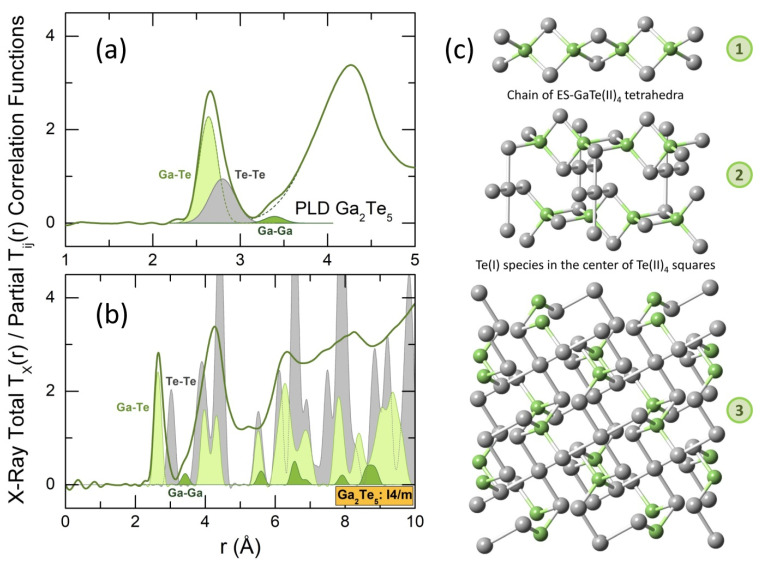
Diffraction data in r-space: the X-ray total correlation function $T_{X}\left(r\right)$ for g-Ga_2_Te_5_ PLD film, showing (**a**) a four-peak Gaussian fitting of the nearest (NN) and second (2N) neighbor features between 2.4 and 4.5 Å; the Ga-Te and Te-Te NNs are highlighted in light green and gray, respectively; the Ga-Ga 2Ns are green; (**b**) a comparison with the partial correlation functions $T_{\mathrm{ij}}\left(r\right)$ for tetragonal Ga_2_Te_5_ (space group I4/m [26]), derived using the XTAL code [76]; (**c**) the crystal structure of tetragonal gallium pentatelluride, revealing (1) infinite chain of edge-sharing ES-GaTe(II)_4_ tetrahedra along the c -axis, (2) two neighboring chains connected by Te(I) species located in the center of Te(II) squares, (3) an approximate (a,b) projection of the crystal structure.

**Figure 7 nanomaterials-13-02137-f007:**
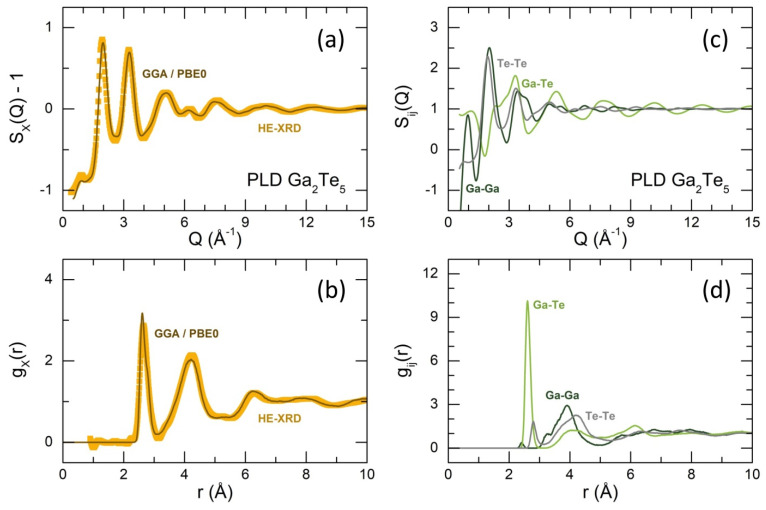
First-principles molecular dynamics modeling of Ga_2_Te_5_ PLD film using GGA/PBE0 approximation; calculated and experimental (**a**) X-ray structure factors $S_{X}\left(Q\right)$; (**b**) pair-distribution functions $g_{X}\left(r\right)$; (**c**) partial structure factors $S_{\mathrm{ij}}\left(Q\right)$; (**d**) partial pair distribution functions $g_{\mathrm{ij}}\left(r\right)$. The Ga−Ga, Ga−Te, and Te−Te partials are dark green, light green, and grey, respectively.

**Figure 8 nanomaterials-13-02137-f008:**
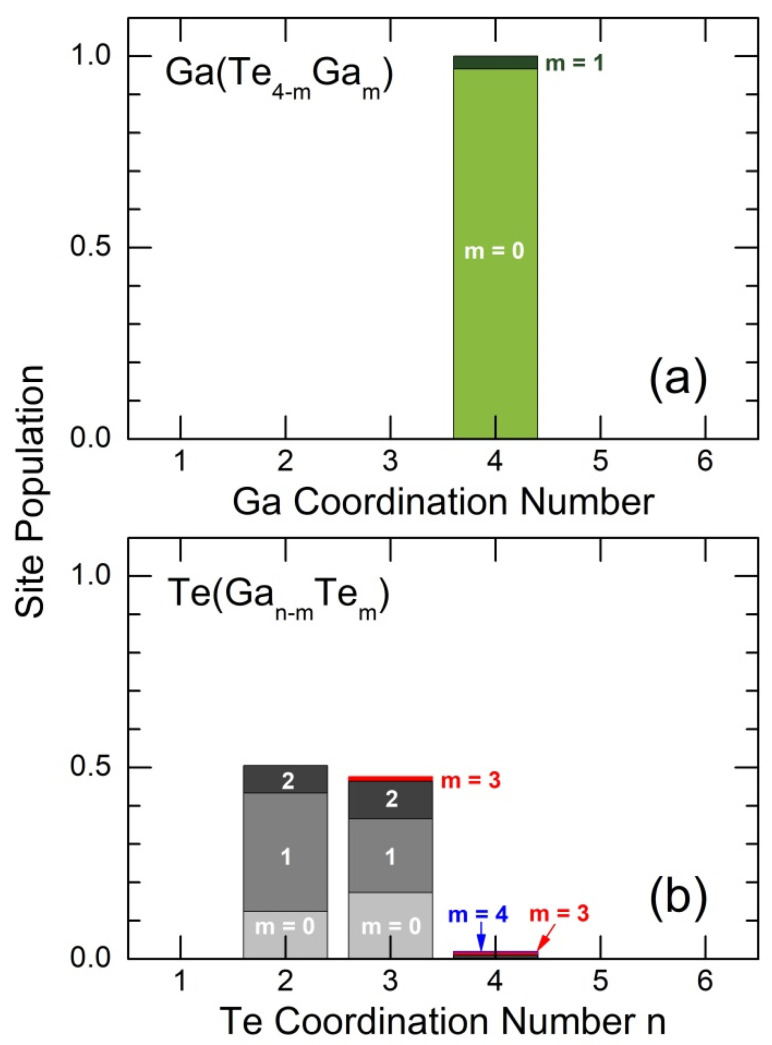
(**a**) Gallium and (**b**) tellurium coordination distributions. Tetrahedral gallium coordination $\mathrm{Ga}\left({\mathrm{Te}}_{4-m}{\mathrm{Ga}}_{m}\right)$ contains a negligible number of Ga-Ga homopolar pairs (m = 1). Multiple tellurium coordination environments $\mathrm{Te}\left({\mathrm{Ga}}_{n-m}{\mathrm{Te}}_{m}\right)$, 1 ≤ n ≤ 4, contain a significant number of Te−Te bonds, 0 ≤ m ≤ 4. See the text for further details.

**Figure 9 nanomaterials-13-02137-f009:**
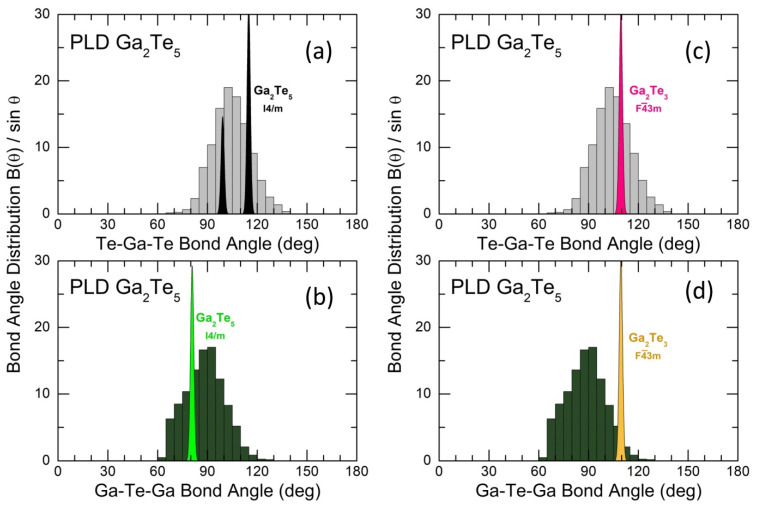
Bond angle distributions B(θ)/sinθ in simulated g-Ga_2_Te_5_: (**a**,**c**) the Te-Ga-Te and (**b**,**d**) Ga-Te-Ga bond angles in comparison with (**a**,**b**) tetragonal Ga_2_Te_5_ and (**c**,**d**) cubic Ga_2_Te_3_.

**Figure 10 nanomaterials-13-02137-f010:**
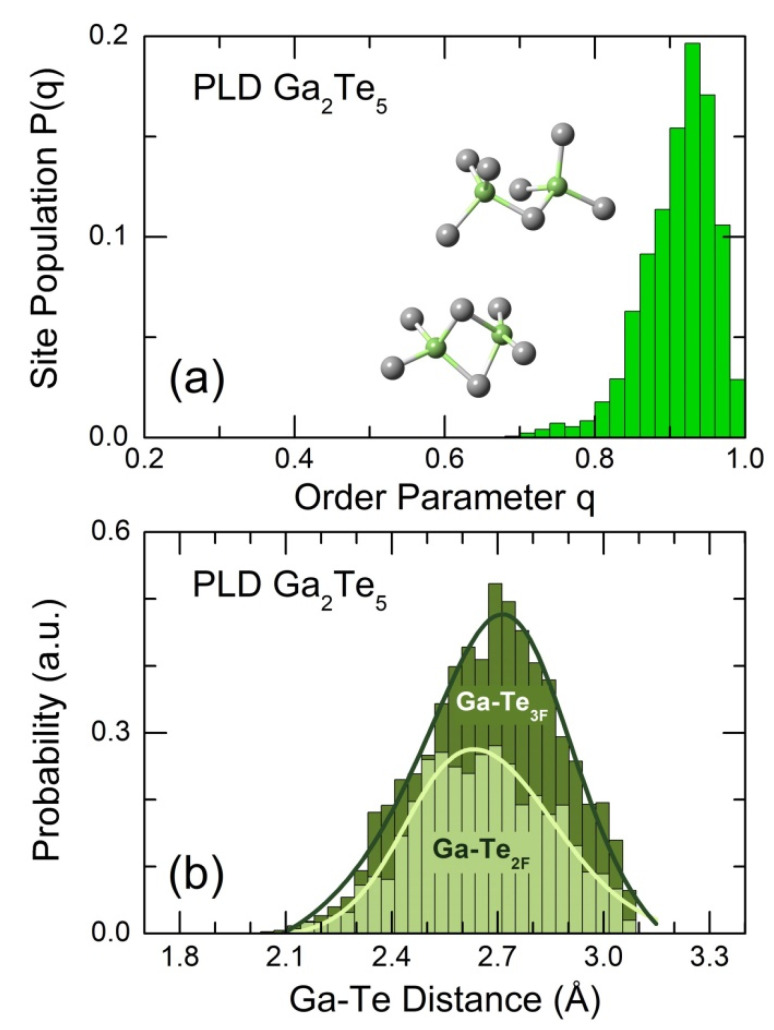
(**a**) Orientational order parameter q [81,82] for 4-fold coordinated Ga species in g-Ga_2_Te_5_ and (**b**) Ga-Te NN distance distributions for two-fold Te_2F_ and three-fold Te_3F_ coordinated tellurium atoms.

**Figure 11 nanomaterials-13-02137-f011:**
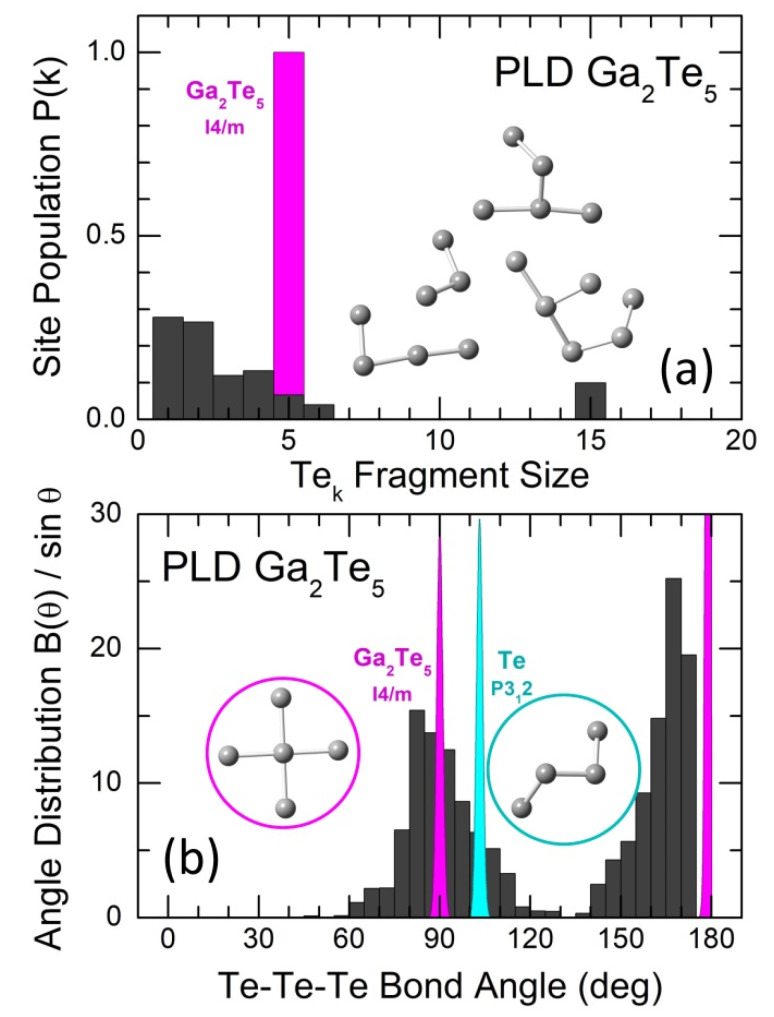
(**a**) Size and (**b**) Te-Te-Te bond angle distribution in ${\mathrm{Te}}_{k}$ oligomeric fragments, k ≤ 15. The only population of 5-membered Te_5_ square-planar fragments (crosses) in tetragonal Ga_2_Te_5_ is also shown in (**a**), as well as the characteristic $B_{\mathrm{TeTeTe}}\left(\theta \right)/\sin\theta$ distributions in trigonal tellurium [57] and tetragonal Ga_2_Te_5_ [26]. Typical ${\mathrm{Te}}_{k}$ fragments in glassy and crystalline pentatellurides are shown in the insets, as well as a part of the helical tellurium chain in c-Te, $P3_{1}2$ [57].

**Figure 12 nanomaterials-13-02137-f012:**
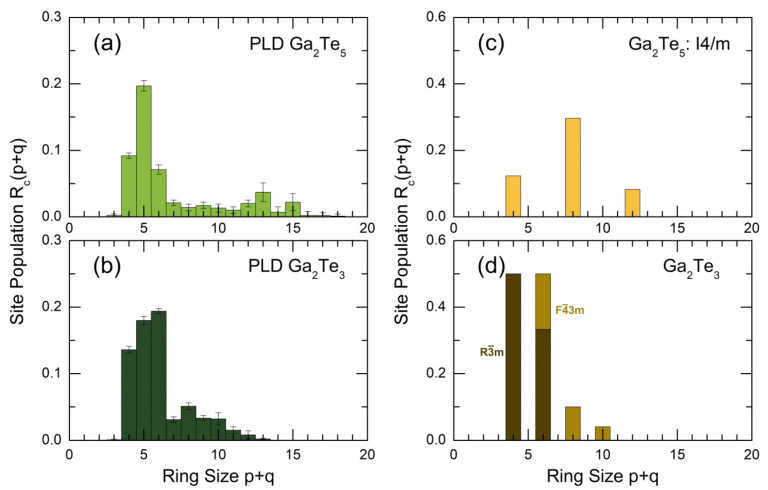
Ring population $R_{c}\left(p\;+q\right)$ [54] in glassy (**a**) Ga_2_Te_5_ (this work), (**b**) Ga_2_Te_3_ [16] PLD films, and crystalline references: (**c**) tetragonal Ga_2_Te_5_ [26], (**d**) cubic [17] and rhombohedral [89] Ga_2_Te_3_.

**Figure 13 nanomaterials-13-02137-f013:**
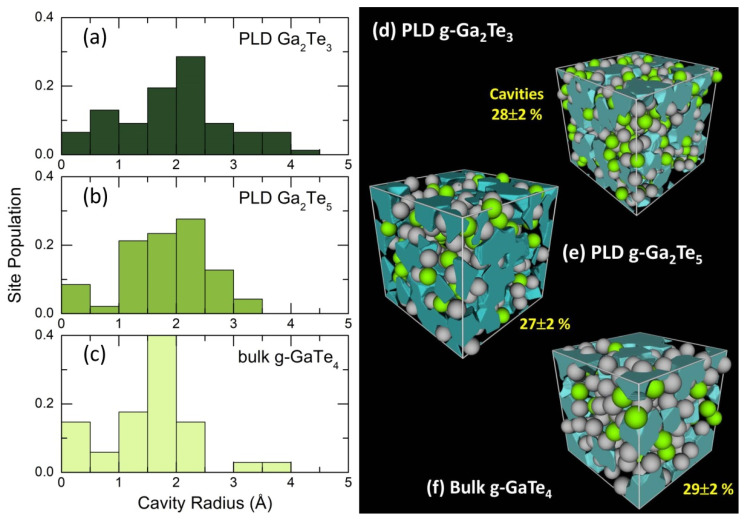
Microscopic voids in glassy Ga-Te alloys; typical distributions of characteristic cavity radii in (**a**) Ga_2_Te_3_, (**b**) Ga_2_Te_5_, and (**c**) GaTe_4_; snapshots of simulation boxes with microscopic voids for (**d**) Ga_2_Te_3_ [16], (**e**) Ga_2_Te_5_ (this work), and (**f**) GaTe_4_ [15], calculated using the Dirichlet–Voronoi approximation [56].

**Figure 14 nanomaterials-13-02137-f014:**
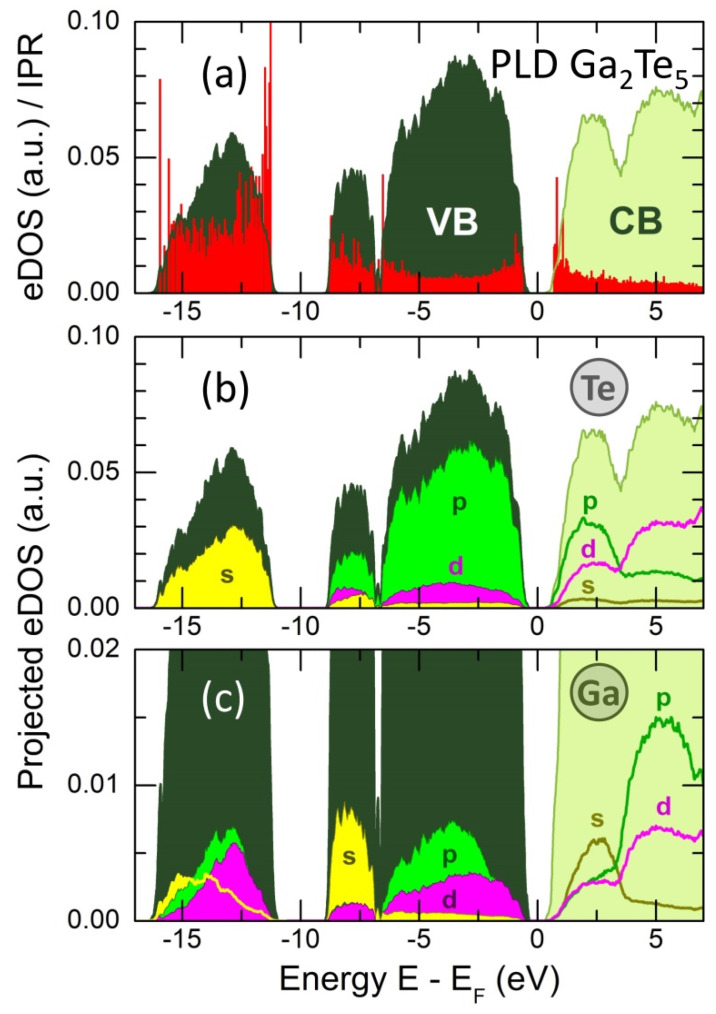
(**a**) Electronic density of states (eDOS) and inverse participation ratio (IPR, red spikes) in g-Ga_2_Te_5_ PLD film at 300 K, and projected DOS on (**b**) tellurium and (**c**) gallium atomic pseudo-wave functions. The s (yellow), p (green) and d (magenta) contributions are shown. VB = valence band; CB = conduction band.

**Table 1 nanomaterials-13-02137-t001:** Interatomic distances $r_{\mathrm{ij}}$ and partial coordination numbers $N_{\mathrm{ij}}$ of the nearest neighbors in glassy Ga_2_Te_5_ PLD films according to high-energy X-ray diffraction and first-principles molecular dynamics.

Ga-Ga		Ga-Te		Te-Te		$N_{Ga-X}$
$r_{ij}\;(Å)$	$N_{ij}$	$r_{ij}\;(Å)$	$N_{ij}$	$r_{ij}\;(Å)$	$N_{ij}$	
High-energy X-ray diffraction						
-	-	2.637(3)	3.98(5)	2.796(5)	1.01(8)	3.98(5)
First-principles molecular dynamics (GGA/PBE0)						
2.417	0.03	2.615 *	3.97	2.802 *	0.93	4.00

* asymmetric peak.

## Data Availability

The data presented in this study are available on request from the corresponding author.

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
