# Peer review of "Decoding the Atomic Structure of Ga2Te5 Pulsed Laser Deposition Films for Memory Applications Using Diffraction and First-Principles Simulations"

_nanomaterials, 2023, doi:10.3390/nano13142137_

Round 1

Reviewer 1 Report

The authors of the submitted manuscript prepared amorphous and crystalline Ga2Te5 based thin films by using pulsed laser deposition (PLD).  The structure of amorphous Ga2Te5 thin film was investigated by using high-energy X-ray diffraction and first-principles calculations. The result were compared with crystalline Ga2Te5 thin film. The manuscript is well written and rich of interesting experimental and theoretical results. I can recommend the publication of present manuscript in the journal mdpi Nanomaterials. However, some revisions should be done before its publication. Here, my suggestions:

1. Introduction: The citation of publications on PLD growth of chalcogenide-based phase change memory materials is missing and I recommend to add the following publications on this topic to the introduction section: APL Materials 5 (2017) 050701, Appl. Phys. Lett. 105 (2014) 221908, Applied Surface Science 536 (2021) 147959, 2D Materials 8 (2021) 045027.

2. Materials and Methods: Provide more details on PLD growth of Ga2Te5 thin films e.g. laser pulse energy and repetition rate.

3. Results: Theoretical and experimental results on atomic structure of amorphous aand crystalline Ga2Te5 should be compared with the Ge2Sb2Te5 and Sb2Te3 phase change alloys with appropriate discussion.

Reviewer 2 Report

This manuscript describes a complete treatise of PLD g-Ga2Te5, from sample preparation to a full characterisation. I specifically wish to emphasise the excellent agreement between FPMD simulations and diffraction experiments (Figure 7). I therefore recommend publication, after the authors have addressed the comments below.

-- Figures 9 and 11: bond angle distribution functions are better presented as the distribution of cosines of bond angles, or equivalently, after a division by sine(Theta). In Figure 9, the present display is not very disturbing (although angles around 90 degrees are over-represented). On the other hand, in Figure 11, angles around 180 degrees may be significant -- and the present display, in terms of bond angles, under-represent angles around 180 degrees. (If the authors don!t believe me then they should calculate bond angle distributions for an ideal gas -- and they would notice that 180 degree angles were missing, whereas there is a peak at 90 degrees, which is not correct.)

--Just below eq. (14), lines 561-563: the line break is not necessary.
